# Bounds on the Probability of Undetected Error for *q*-Ary Codes

**DOI:** 10.3390/e25091349

**Published:** 2023-09-17

**Authors:** Xuan Wang, Huizhou Liu, Patrick Solé

**Affiliations:** 1School of Mathematical Sciences, Anhui University, Hefei 230601, China; wang_xuan_ah@163.com; 2State Grid Anhui Electric Power Co., Ltd., Hefei 230601, China; 18756027866@163.com; 3I2M, CNRS, Aix-Marseille Univetsity, Centrale Marseille, 13009 Marseilles, France

**Keywords:** error correcting codes, probability of undetected error, linear programmming

## Abstract

We study the probability of an undetected error for general *q*-ary codes. We give upper and lower bounds on this quantity, by the Linear Programming and the Polynomial methods, as a function of the length, size, and minimum distance. Sharper bounds are obtained in the important special case of binary Hamming codes. Finally, several examples are given to illustrate the results of this paper.

## 1. Introduction

Let $A=\{a_{1},\ldots ,a_{q}\}$ be an *alphabet* with *q* distinct symbols, where q⩾2 and the alphabet do not have any structure. For instance, *A* can be $F_{q}$, the finite field with *q* elements, or $Z_{q}$, the ring of integers modulo *q*. Moreover, a linear [n,k] code is a subspace of the vector space $F_{q}^{n}$ and *k* is the dimension of the subspace. For every two vectors x, $y\in A^{n}$, the (Hamming) distance $d_{H}\left(x,y\right)$ between x and y is defined as the number of coordinates where they are different. A nonempty subset *C* of $A^{n}$ with cardinality *M* is called a *q*-ary (n,M) code, whose elements are called *codewords*. The minimum distance *d* of the code *C* is the minimum distance between any two different codewords in *C*. The distance distribution of *C* is defined as
(1)$$A_{i}=\frac{1}{M}\left|\left\{\left(x,y\right):x,y\in C,d_{H}\left(x,y\right)=i\right\}\right|,\;i=0,\;1,\ldots ,\;n.$$

Assume that the code *C* is used for error detection on a discrete memoryless channel with *q* inputs and *q* outputs. Each symbol transmitted has a probability 1−p of being received correctly and a probability $p_{q}=p/\left(q-1\right)$ of being transformed into each of the q−1 other symbols. It is natural to let 0⩽p⩽(q−1)/q. Such a channel model is called a *q*-ary symmetric channel qSC(p). When such a code is used on the symmetric *q*-ary channel qSC(p), errors occur with a probability $\frac{p}{q-1}$ per symbol.

Let x∈C be the codeword transmitted and $y=x+e\in F_{q}^{n}$ be the vector received, where e=y−x is the error vector from the channel noise. Obviously, e∈C if and only if y∈C. Note that the decoder will accept y as error free if y∈C. Clearly, this decision is wrong, and such an error is not detected. Thus, when error detection is being used, the decoder will make a mistake and accept a codeword which is not the one transmitted if and only if the error vector is a nonzero codeword [1,2]. In this way, the probability that the decoder fails to detect the existence of an error is called the probability of undetected error and denoted by $P_{ue}\left(C,p\right)$, which is defined as
(2)$$P_{ue}\left(C,p\right)=\sum_{j=1}^{n}A_{j}{\left(\frac{p}{q-1}\right)}^{j}{\left(1-p\right)}^{n-j}.$$
In general, the smaller the probability of undetected error $P_{ue}$ for some *p*, the better the code performs in error detection. However, this function is difficult to characterize in general.

As for the code *C*, comparing its $P_{ue}$ with the average probability $\bar{P_{ue}}$ [3,4] for the ensemble of all *q*-ary linear [n,k] codes is a natural way to decide whether *C* is suitable for error detection or not, where
$$\bar{P_{ue}}\left(p\right)=q^{-(n-k)}\left(1-{\left(1-p\right)}^{k}\right).$$
According to [4], there exists a code *C* such that $P_{ue}\left(C,p\right)>q^{-(n-k)}$ and there are many codes, the $P_{ue}$ of each of whom is smaller than $q^{-(n-k)}$. In fact, it was commonly assumed that $P_{ue}\left(C,p\right)⩽q^{-(n-k)}$ for the linear [n,k] code *C* in [5], where $q^{-(n-k)}=q^{-r}$ is called the $q^{-r}$ bound. The $q^{-r}$ bound is satisfied for certain specific codes, e.g., Hamming codes and binary perfect codes, when 0<p<1/2.

For the worst channel condition, i.e., when p=(q−1)/q,
$$P_{ue}\left(C,\frac{q-1}{q}\right)=q^{-(n-k)}\left(1-{\left(1-\frac{q-1}{q}\right)}^{k}\right)=\bar{P_{ue}}\left(\frac{q-1}{q}\right).$$
From the above formula, a code *C* is called *good* if $P_{ue}\left(C,p\right)⩽P_{ue}\left(\left(q-1\right)/q\right)$ for all 0<p<(q−1)/q. In particular, if $P_{ue}\left(C,p\right)$ is an increasing function of *p* in the interval [0,(q−1)/q], then the code is good, and the code is called *proper*. There are many proper codes [1], for example, perfect codes (and their extended codes and their dual codes), primitive binary 2-error correcting BCH codes, a class of punctured of Simplex codes, MDS codes, and near MDS codes (see [5,6,7,8,9] for details). Moreover, for practical purposes, a *good* binary code *C* may be defined a bit different, i.e., $P_{ue}\left(C,p\right)⩽cP_{ue}\left(C,1/2\right)$ for every 0⩽p⩽1/2 and a reasonably small c⩾1. Furthermore, an infinite class C of binary codes is called *uniformly good* if there exists a constant *c* such that for every 0⩽p⩽1/2 and C∈C, the inequality $P_{ue}\left(C,p\right)⩽cP_{ue}\left(C,1/2\right)$ holds. Otherwise, it is called *ugly*, for example, some special Reed–Muller codes are ugly (see [10]).

Another way to assess the performance of a code for error detection is to give bounds of the probability of undetected error. In [11], Abdel-Ghaffar defined the *combinatorial invariant* $F_{j}$ of the code *C* and proved that
$$P_{ue}\left(C,p\right)=\sum_{j=1}^{n}F_{j}{\left(\frac{p}{q-1}\right)}^{j}{\left(1-\frac{qp}{q-1}\right)}^{n-j},$$
where
Fj=∑i=1jAin−in−j,j=1, 2,…, n.
Using combinatorial arguments, Abdel-Ghaffar [11] obtained a lower bound on the undetected error probability $P_{ue}\left(C,p\right)$. Later, Ashikhmin and Barg called $F_{j}$ the binomial moments of the distance function and derived more bounds for $P_{ue}$ (see [12,13]).

In particular, constant weight codes are attractive and many bounds are developed, for example, binary constant weight codes (see [14,15]) and *q*-ary constant weight codes (see [16]). In fact, the probability of an undetected error for binary constant weight codes has been studied and can be given explicitly (see [14,16]).

Note that when $A=F_{q}$ and p→0, according to Equation (Equation 2), we have
(3)$$P_{ue}\left(C,p\right)∼A_{d}{p_{q}}^{d}{\left(1-p\right)}^{n-d},$$
where $p_{q}=p/\left(q-1\right)$, *d* is the minimum distance of *C* and $A_{d}$ is called the *kissing number* of the linear code *C*. In 2021, Solé et al. [17] studied the kissing number by Linear Programming and the Polynomial Method. They gave bounds for $A_{d}$ under different conditions and made tables for some special parameters. Motivated by the work, this paper is devoted to studying the function $P_{ue}$ using the same techniques.

The rest of this paper is organized as follows. In Section 2, we briefly give the definition of the (dual) distance distribution of *q*-ary codes and give some trivial bounds of the probability of an undetected error. In Section 3.1, linear programming bounds are discussed. The applications of Krawtchouk polynomial (Polynomial Method) to error detection are given in Section 3.2. In Section 4, some bounds better than the $2^{-m}$ bound are given for binary Hamming codes. Finally, we end with some concluding remarks in Section 5.

## 2. Preliminaries

Recall some basic definitions and notations from [2,18,19,20]. Throughout this paper, to simplify some formulas, we let $p_{q}=\frac{p}{q-1}$ and $k=\log_{q}\left|C\right|$ for some real *k*. Furthermore, in this paper, it is natural to define p<(q−1)(1−p), equivalently, $p_{q}<1-p$.

### 2.1. Dual Distance Distribution

Assume that $A=F_{q}$ is the finite field of size *q* and *C* is a subspace of $F_{q}^{n}$, i.e., *C* is a linear code over $F_{q}$. Then, the *dual code* $C^{⊥}$ of *C* is the orthogonal complement of the subspace *C*. That is to say,
$$C^{⊥}=\left\{v\in F_{q}^{n}:v\cdot u=0\;for\;all\;u\in C\right\},$$
where $v\cdot u=\sum_{i=1}^{n}v_{i}u_{i}$, $u=(u_{1},\ldots ,u_{n})$ and $v=(v_{1},\ldots ,v_{n})$. The distance distribution $A_{i}^{\prime }$ of $C^{⊥}$ can be determined similarly. It is well known (see Chapter 5. §2. in [2]) that
(4)$$A_{i}^{\prime }=\frac{1}{\left|C\right|}\sum_{i=0}^{n}A_{j}P_{i}\left(j\right),$$
where $P_{i}\left(j\right)$ denotes the Krawtchouk polynomial of degree *i*. For each integer q⩾2, the *Krawtchouk polynomial*$P_{k}\left(x;n\right)$ is defined as
Pk(x;n)=∑j=0k(−1)jxjn−xk−j(q−1)k−j.
When there is no ambiguity for *n*, the function $P_{k}\left(x;n\right)$ is often simplified to $P_{k}\left(x\right)$.

Note that Equation (Equation 4) holds when *C* is linear. When *C* is nonlinear, the *dual distance distribution* $A_{i}^{\prime }$ is defined by Equation (Equation 4). Furthermore, by the MacWilliams–Delsarte inequality,
(5)$$A_{i}^{\prime }⩾0,$$
holds for all i=0, 1,⋯, n. Moreover, $A_{0}=1$ and
(6)$$q^{k}=1+\sum_{j=1}^{n}A_{j},\;\mathrm{when}\;\left|C\right|=q^{k}.$$

### 2.2. Probability of Undetected Error

The *q-ary symmetric channel* with symbol probability *p*, where 0⩽p⩽(q−1)/q, is defined as follows: symbols from some alphabet *A* with *q* elements are transmitted over the channel, and
$$P\left(b\;\mathrm{received}|a\;\mathrm{sent}\right)=\left\{\begin{array}{cc} 1-p, & b=a,\\ \frac{p}{q-1}, & b\neq a, \end{array}\right.$$
where P(breceived∣asent) is the conditional probability that *b* is received, given that *a* is sent. For a *q*-ary code *C*, when it is used on such a channel, it is possible that the decoder fails to detect the existence of the errors. Thus, $P_{ue}$, the function in terms of the weight distribution of *C* is given in Equation (Equation 2). Clearly, this is a difficult computational problem for large parameters *n*, *k*, *d*, and *q* (see [2]). Hence, it is better to give bounds for $P_{ue}$. For example, here are some trivial bounds.

**Theorem** **1.**
*For every q-ary code C with $\left|C\right|=q^{k}$, if p<(q−1)(1−p), then*

$$\left(q^{k}-1\right){p_{q}}^{n}⩽P_{ue}\left(C,p\right)⩽\left(q^{k}-1\right){p_{q}}^{d}{\left(1-p\right)}^{n-d},$$

*where $p_{q}=\frac{p}{q-1}$. Especially, when q=2 and $0<p<\frac{1}{2}$, we have*

$$\left(2^{k}-1\right)p^{n}⩽P_{ue}\left(C,p\right)⩽\left(2^{k}-1\right)p^{d}{\left(1-p\right)}^{n-d}.$$



**Proof.** It is easy to check that ${p_{q}}^{j}{\left(1-p\right)}^{n-j}>{p_{q}}^{j+1}{\left(1-p\right)}^{n-j-1}$ if and only if p<(q−1)(1−p). Hence,
$$P_{ue}=\sum_{j=d}^{n}A_{j}{p_{q}}^{j}{\left(1-p\right)}^{n-j}⩽{p_{q}}^{d}{\left(1-p\right)}^{n-d}\sum_{j=d}^{n}A_{j}=\left(q^{k}-1\right){p_{q}}^{d}{\left(1-p\right)}^{n-d},$$
since ${p_{q}}^{j}{\left(1-p\right)}^{n-j}⩽{p_{q}}^{d}{\left(1-p\right)}^{n-d}$ when j⩾d. The lower bound can be obtained similarly. □

The above bounds are trivial. However, they are both tight, because simplex codes over the finite field $F_{q}$ attain these bounds.

### 2.3. Some Special Bounds

It is clear that the general bounds given by Theorem 1 will be much larger (or smaller) than the true value of $P_{ue}$ for a fixed code. If the distance distribution is known, one computes $P_{ue}\left(C,p\right)$ (as a function of *p*), and if we know some particular information about the distance distribution, then we may get some bounds. The following is a special case and more thoughts can be seen in Section 4.

**Theorem** **2.**
*Let C be a binary code with $A_{n}=1$ and $A_{i}=A_{n-i}$ for 1⩽i⩽n−1, then*

(7)
$$P_{ue}\;=\;\left\{\begin{array}{cc} p^{n}\;+\;\sum_{j=d}^{t}A_{j}\left(p^{j}{\left(1-p\right)}^{n-j}\;+\;p^{n-j}{\left(1-p\right)}^{j}\right), & n\;=\;2t\;+\;1,\\ p^{n}\;+\;A_{t}p^{t}{\left(1-p\right)}^{t}\;+\;\sum_{j=d}^{t-1}A_{j}\left(p^{j}{\left(1-p\right)}^{n-j}+p^{n-j}{\left(1-p\right)}^{j}\right), & n\;=\;2t. \end{array}\right.$$

*Moreover, when d⩽t, we have*

$$P_{ue}\;⩽\;\left\{\begin{array}{cc} p^{n}\;+\;\left(2^{k-1}\;-\;1\right)\left(p^{d}{\left(1-p\right)}^{n-d}\;+\;p^{t+1}{\left(1\;-\;p\right)}^{t}\right), & n\;=\;2t\;+\;1,\\ p^{n}\;+\;A_{t}p^{t}{\left(1\;-\;p\right)}^{t}\;+\;\left(2^{k-1}\;-\;\frac{A_{t}}{2}\;-\;1\right)\left(p^{d}{\left(1\;-\;p\right)}^{n-d}\;+\;p^{t+1}{\left(1\;-\;p\right)}^{t-1}\right), & n\;=\;2t, \end{array}\right.$$

*and*

$$P_{ue}\;⩾\;\left\{\begin{array}{cc} p^{n}\;+\;\left(2^{k-1}\;-\;1\right)\left(p^{t}{\left(1-p\right)}^{t+1}\;+\;p^{n-d}{\left(1-p\right)}^{d}\right), & n\;=\;2t\;+\;1,\\ p^{n}\;+\;A_{t}p^{t}{\left(1\;-\;p\right)}^{t}\;+\;\left(2^{k-1}\;-\;\frac{A_{t}}{2}\;-\;1\right)\left(p^{t-1}{\left(1\;-\;p\right)}^{t+1}\;+\;p^{n-d}{\left(1\;-\;p\right)}^{d}\right), & n\;=\;2t, \end{array}\right.$$

*where $0<p<\frac{1}{2}$ and d⩽t.*


**Proof.** By the definition of $P_{ue}$, Equation (Equation 7) holds if $A_{i}=A_{n-i}$ and $A_{n}=1$. Due to $0<p<\frac{1}{2}$, It is easy to check that $p^{n-j}{\left(1-p\right)}^{j}⩽p^{j}{\left(1-p\right)}^{n-j}$, where 0⩽j⩽⌊n/2⌋. In addition, if n=2t+1, then $\sum_{j=d}^{t}A_{j}=\left(2^{k}-2\right)/2=2^{k-1}-1$. Similarly for the case n=2t. Hence, we get the bounds. □

**Remark** **1.**
*If the binary code C satisfies $A_{i}=A_{n-i}$ and $A_{n}=0$, we can get the following bounds:*

$$P_{ue}\;⩽\;\left\{\begin{array}{cc} 2^{k-1}\left(p^{d}{\left(1\;-\;p\right)}^{n-d}\;+\;p^{t+1}{\left(1\;-\;p\right)}^{t}\right), & n\;=\;2t\;+\;1,\\ A_{t}p^{t}{\left(1\;-\;p\right)}^{t}\;+\;\left(2^{k-1}\;-\;\frac{A_{t}\;+\;A_{0}}{2}\right)\left(p^{d}{\left(1\;-\;p\right)}^{n-d}\;+\;p^{t+1}{\left(1\;-\;p\right)}^{t-1}\right), & n\;=2t, \end{array}\right.$$

*and*

$$P_{ue}\;⩾\;\left\{\begin{array}{cc} 2^{k-1}\left(p^{t}{\left(1\;-\;p\right)}^{t+1}\;+\;p^{n-d}{\left(1\;-\;p\right)}^{d}\right), & n\;=\;2t\;+\;1,\\ A_{t}p^{t}{\left(1\;-\;p\right)}^{t}\;+\;\left(2^{k-1}\;-\;\frac{A_{t}\;+\;A_{0}}{2}\right)\left(p^{t-1}{\left(1\;-\;p\right)}^{t+1}\;+\;p^{n-d}{\left(1\;-\;p\right)}^{d}\right), & n\;=\;2t. \end{array}\right.$$

*Here, 0, the all zero vector, may not be a codeword.*


**Example** **1.**
*For a binary linear code, if the all-one vector 1 is a codeword, then $A_{i}=A_{n-i}$. So, Theorem 2 can be applied to many codes, for example, Hamming codes. It is known that the binary Hamming code $H_{m}$ is a linear $\left[n=2^{m}-1,k=2^{n}-1-m,3\right]$ code. The distance distribution of the [15,11,3] Hamming code $H_{4}$ is listed in Table 1. According to Theorem 2, the values of the bounds and true probability can be seen in Figure 1.*


## 3. Universal Bounds for *q*-Ary Codes

In this section, we will discuss the bounds for $P_{ue}$ using different methods. These bounds are for general codes, thus they do not look so good. Meanwhile, compared with some known bounds, they do not perform better. However, it is the first as far as we know to give bounds for $P_{ue}$ using the following two methods, though they have been shown in [21,22] due to different thoughts.

### 3.1. Linear Programming Bounds

Consider the linear programming problem M(n,k,d,p) that maximizes the objec- tive function
$$\sum_{j=1}^{n}A_{j}{p_{q}}^{j}{\left(1-p\right)}^{n-j}$$
under the constraints:(1)$A_{j}⩾0,$(2)$\sum_{j=1}^{n}A_{j}=q^{k}-1$,(3)$\sum_{j=1}^{n}A_{j}P_{i}\left(j\right)⩾-P_{i}\left(0\right)$,(4)$A_{1}=A_{2}=\cdots =A_{d-1}=0$.

Likewise, let m(n,k,d,p) be the minimization of the same objective function under the same constraints.

**Theorem** **3.**
*If C is a q-ary code of parameters $(n,q^{k},d),$ then $m\left(n,k,d,p\right)⩽P_{ue}⩽M\left(n,k,d,p\right).$*


**Proof.** The objective function expression comes from (2). Constraint (1) is immediate by the definition of the distance distribution. Constraints (2) and (3) come from Equation (Equation 6) and Equation (Equation 5), respectively. Constraint (4) is a consequence of the definition of minimum distance. □

**Remark** **2.**
*Let f(x) and g(x) be two functions of x, then f≲g if f<g or f∼g, when x→0, where 0<x<1. For example, let $f\left(x\right)=x^{2}+x$ and $g\left(x\right)=x^{3}+x$, then f(x)>g(x) when 0<x<1. But f(x)∼g(x), then f(x)≲g(x) when 0<x<1 and x→0.*


Motivated by Equation (Equation 3) and [17], we have the following result.

**Theorem** **4.**
*Let C be a q-ary ${\left[n,k,d\right]}_{q}$ linear code, then when p→0,*

(8)
$$\left(q^{k}-1-\left\lfloor L\right\rfloor \right){p_{q}}^{d}{\left(1-p\right)}^{n-d}⩽P_{ue}\left(C,p\right)≲\left(q^{k}-1-\left\lceil S\right\rceil \right){p_{q}}^{d}{\left(1-p\right)}^{n-d},$$

*where L (resp. S) denotes the maximum (resp. minimum) of $\sum_{j=d+1}^{n}A_{j}$ subject to the 2n−d constraints*

$$-P_{i}\left(0\right)-\left(q^{k}-1\right)P_{i}\left(d\right)⩽\sum_{j=d+1}^{n}A_{j}\left(P_{i}\left(j\right)-P_{i}\left(d\right)\right),$$

*for i=1, 2,…, n and j=d+1, d+2,…, n.*


**Proof.** It is clear that $P_{ue}\left(C,p\right)⩾A^{d}{p_{q}}^{d}{\left(1-p\right)}^{n-d}$, then by [17], we get the left side of Equation (Equation 8). As for the right side, if $A_{d}<q^{k}-1-\left\lceil S\right\rceil$ and *p* is small enough, then by Equation (Equation 3), $P_{ue}\left(C,p\right)<\left(q^{k}-1-\left\lceil S\right\rceil \right){p_{q}}^{d}{\left(1-p\right)}^{n-d}$. Otherwise, $A_{d}=q^{k}-1-\left\lceil S\right\rceil$ and then, $P_{ue}\left(C,p\right)∼\left(q^{k}-1-\left\lceil S\right\rceil \right){p_{q}}^{d}{\left(1-p\right)}^{n-d}$. □

Table 2 is a part of Table I in [17], which is helpful to give bounds for $P_{ue}$.

**Example** **2.**
*Let $C_{1}$ be a binary [15,4,8] code, then*

$$P_{ue}\left(C_{1},p\right)∼15p^{8}{\left(1-p\right)}^{7}.$$

*As for the binary [12,4,6] code $C_{2}$, we have*

$$11p^{6}{\left(1-p\right)}^{6}<P_{ue}\left(C_{2},p\right)<14p^{6}{\left(1-p\right)}^{6}.$$

*Obviously, for any [n,k,d] code, one can give bounds for its $P_{ue}$.*


**Remark** **3.**
*From the above discussion, it is clear that our bounds depend solely on the three parameters [n,k,d] of the code, and [n,k,d] is the minimal requirement to use a code in practice.*


### 3.2. Polynomial Method

In this section, we will give some general bounds for $P_{ue}$ for any binary $\left(n,2^{k},d\right)$ code. Recall the definition of the Krawtchouk polynomials and some properties. The following identity is a Polynomial Method of expressing the duality of LP.

**Lemma** **1.**
*Let β(x)∈Q[x] be the polynomial whose Krawtchouk expansion is*

$$\beta \left(x\right)=\sum_{j=0}^{n}\beta_{j}P_{j}\left(x\right).$$

*Then we have the following identity*

(9)
$$\sum_{i=0}^{n}\beta \left(i\right)A_{i}=q^{k}\sum_{j=0}^{n}\beta_{j}A_{j}^{\prime }.$$



**Proof.** Immediate by Equation (Equation 4), upon swapping the order of summation. □

From now on, we denote the coefficient of Krawtchouk expansion of the polynomial f(x) of degree *n* by $f_{j}$, j=0, 1,⋯, n, i.e., $f\left(x\right)=\sum_{j=0}^{n}f_{j}P_{j}\left(x\right)$.

The first main result of this section is inspired by Theorem 1 in [23], and given as follows.

**Theorem** **5.**
*Let β(x) and γ(x) be polynomials over Q such that $\beta_{j}⩽0$, $\gamma_{j}⩾0$ for j⩾1 and $\gamma \left(i\right)⩽{p_{q}}^{i}{\left(1-p\right)}^{n-i}⩽\beta \left(i\right)$ for all i with $A_{i}\neq 0$. Then we have the upper bound*

(10)
$$P_{ue}⩽q^{k}\beta_{0}-\beta \left(0\right),$$

*and the lower bound*

(11)
$$P_{ue}⩾q^{k}\gamma_{0}-\gamma \left(0\right).$$



**Proof.** By Lemma 1, we have
$$\sum_{j=0}^{n}A_{j}\beta \left(j\right)⩽\beta_{0}q^{k}.$$Returning to the definition of $P_{ue}$ and using the property of $\beta \left(j\right)⩾{p_{q}}^{j}{\left(1-p\right)}^{n-j}$, we get
$$P_{ue}=\sum_{j=1}^{n}A_{j}{p_{q}}^{j}{\left(1-p\right)}^{n-j}⩽\sum_{j=1}^{n}A_{j}\beta \left(j\right)⩽q^{k}\beta_{0}-\beta \left(0\right).$$The proof of the lower bound is analogous and ommitted. □

**Remark** **4.**
*The above result is a special case of Proposition 5 in [22]. More general setting of the linear programming bounds from Section 3 (Theorem 5) were already considered in [21,22].*


The following are some properties of the Krawtchouk expansion, and we omit the proof, since they are not difficult.

**Lemma** **2**([24] Corollary 3.13)**.**
*Let $f\left(x\right)=\sum_{j=0}^{n}f_{j}P_{j}\left(x\right)$ and $g\left(x\right)=\sum_{j=0}^{n}g_{j}P_{j}\left(x\right)$ be polynomials over Q, where $f_{j}⩾0,\;g_{j}⩾0$, 0⩽j⩽n. Then the coefficients of the Krawtchouk expansion of λf(x)+μg(x) are nonnegative, where λ,μ are nonnegative rational numbers.*

#### 3.2.1. Upper Bounds

For convenience, let $\delta_{i,j}$ be the Kronecker symbol, i.e.,
$$\delta_{i,j}=\left\{\begin{array}{cc} 1, & \mathrm{if}\;i=j,\\ 0, & \mathrm{if}\;i\neq \;j. \end{array}\right.$$

**Lemma** **3.**
*For general q, the coefficients of the Krawtchouk expansion of the following polynomial*

$$g_{i}\left(x\right)=\frac{{\left(-1\right)}^{i-1}}{(i-1)!(n-i)!}\frac{\prod_{j=1}^{n}\left(j-x\right)}{i-x},$$

*are all nonnegative if and only if i is odd, where 1⩽i⩽n is an integer and 0!=1. Moreover, $g_{i}\left(j\right)=\delta_{i,j}$.*


**Proof.** Let
$$h\left(x\right)=\frac{q^{n-d+1}}{s-x}\prod_{j=d}^{n}\left(1-\frac{x}{j}\right)=\sum_{j=0}^{n}h_{j}P_{j}\left(x\right),$$
where d⩽s⩽n. Then, by Proposition 5.8.2 in [20],
hi=1qn∑j=0nh(j)Pj(i)=1qd−1∑j=0d−1n−jn−d+1Pj(i)s−j/nd−1⩾1qd−1s∑j=0d−1n−jn−d+1Pj(i)/nd−1=1sn−id−1/nd−1⩾0.Note that if d=1, we have
$$h\left(x\right)=\frac{q^{n}}{n!}{\left(-1\right)}^{i-1}\left(i-1\right)!\left(n-i\right)!g_{s}\left(x\right).$$According to Lemma 2, the coefficients of the Krawtchouk expansion of ${\left(-1\right)}^{i-1}g_{i}\left(x\right)$ are all nonnegative.Obviously, for any j≠i, $g_{i}\left(j\right)=0$, because *j* is a root of $g_{i}\left(x\right)$. Moreover,
$$\begin{array}{cc} g_{i}\left(i\right) & =\frac{{\left(-1\right)}^{i-1}}{(i-1)!(n-i)!}\prod_{\ell =1}^{i-1}\left(\ell -i\right)\prod_{\ell =i+1}^{n}\left(\ell -i\right)\\ & =\frac{{\left(-1\right)}^{i-1}}{(i-1)!(n-i)!}\left({\left(-1\right)}^{i-1}\left(i-1\right)!\left(n-i\right)!\right)=1, \end{array}$$
which means $g_{i}\left(j\right)=\delta_{i,j}$. □

**Theorem** **6.**
*Let C be a binary code with the distance distribution $A_{j}$, where $A_{j}=0$ for all possible odd j, then*

(12)
Pue⩽∑evenipi(1−p)n−ini12n−k+1,

*where even i means that i runs through the even intergers between d and n.*


**Proof.** According to Lemma 3, the coefficients of the Krawtchouk expansion of the following polynomial:
$$g_{i}\left(x\right)=\frac{{\left(-1\right)}^{i-1}}{(i-1)!(n-i)!}\frac{\prod_{j=1}^{n}\left(j-x\right)}{i-x}$$
are nonnegative if and only if *i* is odd. Then, let
$$f\left(x\right)=\sum_{\mathrm{even}\;i}p^{i}{\left(1-p\right)}^{n-i}g_{i}\left(x\right)=\sum_{j=0}^{n}f_{j}P_{j}\left(x\right).$$Hence, $f_{j}⩽0$, $f\left(i\right)=p^{i}{\left(1-p\right)}^{n-i}$ for even *i* and f(i)=0 for odd *i*. By the proof of Theorem 5,
$$P_{ue}⩽2^{k}f_{0}-f\left(0\right),$$
where
f(0)=∑eveni(−1)pi(1−p)n−ini,
and
f0=12n∑evenipi(1−p)n−ini.Thus, the upper bound follows from Theorem 5. □

**Remark** **5.**
*If C is linear, then $A_{i}$ is the number of codewords of weight i, which implies that Ai⩽ni. Hence,*

Pue⩽∑i∈Ipi(1−p)n−ini,

*where $I=\{i|A_{i}\neq 0\}$. Moreover, if $A_{i}=0$ for all odd i, then*

(13)
Pue⩽∑evenipi(1−p)n−ini.



**Example** **3.**
*Consider the Nordstrom–Robinson code, it is a binary nonlinear code with the distance distribution in Table 3. Moreover, the weight distribution is the same as the distance distribution. By Equation *(Equation 2)*,*

$$P_{ue}=112p^{6}{\left(1-p\right)}^{10}+30p^{8}{\left(1-p\right)}^{8}+112p^{10}{\left(1-p\right)}^{6}+p^{16}.$$

*According to Theorem 6, the values of the upper bound and true probability can be seen in Figure 2.*


**Example** **4.**
*Let E be the set of binary vectors of length n and even weight, then it is actually the Reed–Muller code RM(n−1,n) in Problem 5 in [2] and is generated by all the binary vectors of weight 2. Hence,*

Pue(E,p)=∑i=1⌊n/2⌋n2ip2i(1−p)n−2i.



**Remark** **6.**
*The bound is suitable for many codes, and thus it seems not good. In fact, there exists some code C, whose $P_{ue}$ is very large.*


Motivated by [17], we have the following upper bounds for linear codes over $F_{2}$.

**Proposition** **1.**
*When C is a q-ary linear [n,k,d] code and p is small enough, we have the following statements:*
*(1)* 
*If n+1+qd−nq>0, then*

$$P_{ue}≲\frac{q^{k}+nq-n-1}{n-nq+1+qd}{p_{q}}^{d}{\left(1-p\right)}^{n-d};$$

*(2)* 
*If n+qd−nq−1<0, then*

$$P_{ue}≲\frac{q^{k-2}n\left(qn-n-qd+1\right)+n\left(d-1\right)}{n-d}{p_{q}}^{d}{\left(1-p\right)}^{n-d};$$

*(3)* 
*If q=2, n−2d>0, ${\left(n-2d+2\right)}^{2}>n$, and $A_{i}\neq 0$ only if d⩽i⩽n−2d, then*

$$P_{ue}≲\frac{2^{k-2}\left({\left(n-2d+2\right)}^{2}-n\right)+\left(d-1\right)\left(n-d+1\right)}{n+1-2d}p^{d}{\left(1-p\right)}^{n-d}.$$




**Proof.** These three bounds can be deduced easily by Equation (Equation 3) and Corollaries 4–6 in [17]. □

**Remark** **7.**
*The results in Corollary 4–6 in [17] are actually the upper bounds of $A_{d}$ under different conditions. Considering Equation *(Equation 3)*, it is necessary to make p small enough. According to the proof of Theorem 4, if $A_{d}$ does not meet such bounds, then “<” holds.*


#### 3.2.2. Lower Bounds

Similar to Proposition 1, by Corollaries 1–3 in [17], we have

**Proposition** **2.**
*If C is a q-ary linear code, then we have the following statements:*
*(1)* 
*If d=⌈(n−1)(q−1)/q⌉, then*

$$P_{ue}⩾\frac{q^{k}-nq+n-1}{(n-d)q-n+1}{p_{q}}^{d}{\left(1-p\right)}^{n-d};$$

*(2)* 
*If qd>nq−n−2q+1, then*

$$P_{ue}⩾\frac{q^{k-2}n\left(n-qn+qd+2q-1\right)-nd-n}{n-d}{p_{q}}^{d}{\left(1-p\right)}^{n-d};$$

*(3)* 
*If q=2 and all weights of C are in [d,n−d], with n−2d>0 and ${\left(n-2d-1\right)}^{2}<n+1$, then*

$$P_{ue}⩾\left(\frac{2^{k-2}\left(n^{2}-4nd-3n\right)+\left(2^{k}+1\right)d\left(d+1\right)}{2d-n}-d-1\right)p^{d}{\left(1-p\right)}^{n-d}.$$




When using quadratic polynomials, we have the following bound.

**Proposition** **3.**
*Let $f_{0}$, $f_{1}$ and $f_{2}$ be nonnegative rational numbers such that*

f0−f1n+f2n2⩽pd(1−p)n−dandf1+nf2⩽2df2,

*then, for a binary $\left(n,2^{k},d\right)$ code, we have*

$$P_{ue}⩾2^{k}f_{0}-p^{d}{\left(1-p\right)}^{n-d}-2f_{1}n,$$

*where $0⩽p⩽\frac{1}{2}$.*


**Proof.** It is known that, when q=2, $P_{0}\left(x\right)=1$, $P_{1}\left(x\right)=n-2x$ and P2(x)=2x2−2nx+n2. Let $f\left(x\right)=f_{0}P_{0}\left(x\right)+f_{1}P_{1}\left(x\right)+f_{2}P_{2}\left(x\right)$ and then it is a quadratic function whose axis of symmetry is $\frac{f_{1}+nf_{2}}{2f_{2}}$. Considering that $p^{i+1}{\left(1-p\right)}^{n-i-1}⩾p^{i}{\left(1-p\right)}^{n-i}$, it is sufficient to show that
$$f\left(n\right)⩽p^{d}{\left(1-p\right)}^{n-d}\;\mathrm{and}\;\frac{f_{1}+nf_{2}}{2f_{2}}⩽d,$$
i.e., $f\left(i\right)⩽f\left(n\right)⩽p^{d}{\left(1-p\right)}^{n-d}⩽p^{i}{\left(1-p\right)}^{n-i}$ for i⩾d. Equivalently,
f0−f1n+f2n2⩽pd(1−p)n−d,f1+nf2⩽2df2.The result follows from Theorem 5. □

## 4. Good Bounds for Hamming Codes

Recall that the *weight enumerator* of the code *C* is the homogeneous polynomial
$$W_{C}\left(x,y\right)=\sum_{c\in C}x^{n-wt(u)}y^{wt(u)},$$
where wt(u) means the Hamming weight the codeword u. The binary Hamming code $H_{m}$ is a $\left[n=2^{m}-1,k=n-m,d=3\right]$ code, with the weight enumerator
$$\frac{{\left(x+y\right)}^{n}+n{\left(x+y\right)}^{(n-1)/2}{\left(x-y\right)}^{(n+1)/2}}{n+1},$$
whose distance distribution $A_{i}$ satisfies
$$\sum_{i=1}^{n}iA_{i}y^{i-1}+\sum_{i=0}^{n}A_{i}y^{i}+\sum_{i=0}^{n-1}\left(n-i\right)A_{i}y^{i+1}={\left(1+y\right)}^{n},$$
and the recurrence $A_{0}=1$, $A_{1}=0$,
(i+1)Ai+1+Ai+(n−i+1)Ai−1=ni.Moreover,
$$\begin{array}{cc} {\left(1+y\right)}^{n} & =\frac{\sum_{i=1}^{n}iA_{i}y^{i}}{y}+\sum_{i=0}^{n}A_{i}y^{i}+ny\sum_{i=0}^{n-1}A_{i}y^{i}-y\sum_{i=0}^{n-1}iA_{i}y^{i}\\ \; & =\sum_{i=1}^{n-1}A_{i}y^{i}\left(\frac{i}{y}-iy\right)+\left(ny+1\right)\sum_{i=1}^{n-1}A_{i}y^{i}+y^{n}+ny^{n-1}+ny+1. \end{array}$$

Let $\alpha \in F_{2^{m}}$ be a primitive element and let $g\left(x\right)\in F_{2}\left[x\right]$ be the minimal polynomial of α with respect to $F_{2}$. According to Exercise 7.20 in [20], g(x) can be regarded as the generator polynomial of a Hamming code. Since deg(g(x))=m>1, then
$$g\left(x\right)\left|\frac{x^{n}-1}{x-1}\right.=1+x+x^{2}+\cdots +x^{n-1},$$
which implies that the all-one vector is a codeword of the Hamming code and $A_{n}=1$.

Note that
$$P_{ue}=\sum_{i=1}^{n}A_{i}p^{i}{\left(1-p\right)}^{n-i}={\left(1-p\right)}^{n}\sum_{i=1}^{n}A_{i}{\left(\frac{p}{1-p}\right)}^{i}.$$Hence,
$$\sum_{i=1}^{n-1}A_{i}{\left(\frac{p}{1-p}\right)}^{i}=\frac{P_{ue}-p^{n}}{{\left(1-p\right)}^{n}}.$$Let $y=\varepsilon =\frac{p}{1-p}$, where p∈(0,1/2), then
$$\begin{array}{cc} \; & \left(n\varepsilon +1\right)\left(P_{ue}-p^{n}\right)+{\left(1-p\right)}^{n}\sum_{i=1}^{n-1}A_{i}\varepsilon^{i}\left(\frac{i}{\varepsilon }-i\varepsilon \right)\\ \; & =1-p^{n}-np{\left(1-p\right)}^{n-1}-np^{n-1}\left(1-p\right)-{\left(1-p\right)}^{n}. \end{array}$$According to Chapter 6, Exercise(E2), page 157 in [2], there are n−4 nonzero weights of $H_{m}$. Considering that $A_{n}=1$, we have $A_{i}=0$ if and only if i=1, 2, n−1, n−2. Since 0<p<1/2, then 0<ε<1 and we have
$$\frac{3}{\varepsilon }-3\varepsilon ⩽\frac{i}{\varepsilon }-i\varepsilon ⩽\frac{n-3}{\varepsilon }-\left(n-3\right)\varepsilon .$$Obviously,
$$\begin{array}{cc} {\left(1-p\right)}^{n}\sum_{i=1}^{n-1}A_{i}\varepsilon^{i}\left(\frac{i}{\varepsilon }-i\varepsilon \right) & ⩽{\left(1-p\right)}^{n}\sum_{i=1}^{n-1}A_{i}\varepsilon^{i}\left(\frac{n-3}{\varepsilon }-\left(n-3\right)\varepsilon \right)\\ \; & =\left(\frac{n-3}{\varepsilon }-\left(n-3\right)\varepsilon \right)\sum_{i=1}^{n-1}A_{i}p^{i}{\left(1-p\right)}^{n-i}\\ \; & =\left(\frac{n-3}{\varepsilon }-\left(n-3\right)\varepsilon \right)\left(P_{ue}-p^{n}\right). \end{array}$$Similarly,
$${\left(1-p\right)}^{n}\sum_{i=1}^{n-1}A_{i}\varepsilon^{i}\left(\frac{i}{\varepsilon }-i\varepsilon \right)⩾\left(\frac{3}{\varepsilon }-3\varepsilon \right)\left(P_{ue}-p^{n}\right).$$Thus,
(14)$$\begin{array}{cc} P_{ue} & ⩽\frac{1-p^{n}-np{\left(1-p\right)}^{n-1}-np^{n-1}\left(1-p\right)-{\left(1-p\right)}^{n}}{\frac{3}{\varepsilon }-3\varepsilon +n\varepsilon +1}+p^{n}\\ \; & =\frac{p\left(1-p\right)-p^{n+1}\left(1-p\right)-np^{2}{\left(1-p\right)}^{n}-np^{n}{\left(1-p\right)}^{2}-p{\left(1-p\right)}^{n+1}}{\left(n-1\right)p^{2}-5p+3}+p^{n} \end{array}$$
and
(15)$$\begin{array}{cc} P_{ue} & ⩾\frac{1-p^{n}-np{\left(1-p\right)}^{n-1}-np^{n-1}\left(1-p\right)-{\left(1-p\right)}^{n}}{\frac{n-3}{\varepsilon }-\left(n-3\right)\varepsilon +n\varepsilon +1}+p^{n}\\ \; & =\frac{p\left(1-p\right)-p^{n+1}\left(1-p\right)-np^{2}{\left(1-p\right)}^{n}-np^{n}{\left(1-p\right)}^{2}-p{\left(1-p\right)}^{n+1}}{\left(n-1\right)p^{2}-\left(2n-7\right)p+n-3}+p^{n}. \end{array}$$

Summarize the above discussions, we get

**Theorem** **7.**
*Let $H_{m}$ be the binary $\left[n=2^{m}-1,k=n-m,3\right]$ Hamming code, then when 0<p<1/2 and m⩾3, we have the upper bound Equation *(Equation 14)* and the lower bound Equation *(Equation 15)* for $P_{ue}$, respectively.*


**Proof.** Note that the upper bound should be larger or equal than the lower bound, then
$$\left(-(2n-7)p+n-3\right)-\left(-5p+3\right)=\left(n-6\right)\left(1-2p\right)⩾0.$$It is sufficient to solve the inequality $n=2^{m}-1>6$, due to 1−2p>0. Hence, m⩾3. □

**Remark** **8.**
*The difference of the upper bound and the lower bound is small.*

*Let $U\left(n,p\right)=H_{1}/H$ and $L\left(n,p\right)=H_{2}/H$ be the bound given by Equation *(Equation 14)* and Equation *(Equation 15)*, respectively, where $H_{1}=\left(n-1\right)p^{2}-5p+3$, $H_{2}=\left(n-1\right)p^{2}-\left(2n-7\right)p+n-3$ and*

$$H=p\left(1-p\right)-p^{n+1}\left(1-p\right)-np^{2}{\left(1-p\right)}^{n}-np^{n}{\left(1-p\right)}^{2}-p{\left(1-p\right)}^{n+1}.$$

*In fact, H is a polynomial of p whose degree n+2 and the leading coefficient is*

$$h_{n+2}=1+{\left(-1\right)}^{n+2}n-n+{\left(-1\right)}^{n+2}=\left(1+{\left(-1\right)}^{n}\right)+n\left({\left(-1\right)}^{n}-1\right)\neq 0,$$

*while the product $H_{1}H_{2}$ is just a polynomial whose degree is *4*. Then,*

$$\begin{array}{cc} U(n,p)-L(n,p) & =\frac{(H_{2}-H_{1})H}{H_{1}H_{2}}=\frac{(n-6)(1-2p)H}{H_{1}H_{2}}\\ \; & ⟶\frac{\left(n-6\right)\left(1-2p\right)h_{n+2}p^{n+2}}{{\left(n-1\right)}^{2}p^{4}}⟶0\;\left(n\to +\infty \right). \end{array}$$

*That is to say, the lower bound and the upper bound are very close. On the other hand,*

$$H_{1}⩾\frac{12n-37}{4(n-1)}\;and\;H_{2}⩾\frac{n+1}{4}.$$

*Then,*

$$\begin{array}{cc} U(n,p)-L(n,p) & =\frac{(H_{2}-H_{1})H}{H_{1}H_{2}}=\frac{(n-6)(1-2p)H}{H_{1}H_{2}}\\ \; & <\frac{\left(n-6)(1-2p)p(1-p\right)}{H_{1}H_{2}}<\frac{\left(n-6)(1-2p)p(1-p\right)}{\frac{12n-37}{4(n-1)}\frac{n+1}{4}}\\ \; & =\frac{16(n-1)(n-6)}{\left(n+1)(12n-37\right)}p\left(1-p\right)\left(1-2p\right)\\ \; & ⩽\frac{\sqrt{3}}{18}\frac{16(n-1)(n-6)}{\left(n+1)(12n-37\right)}⟶\frac{2\sqrt{3}}{27}\approx 0.1283\;\left(n\to +\infty \right). \end{array}$$

*Here, let F(p)=p(1−p)(1−2p), then its derivative is $F^{\prime }\left(p\right)=6p^{2}-6p+1$. Note that the roots of $F^{\prime }\left(p\right)$ are $\frac{3\pm \sqrt{3}}{6}$. Since 0<p<1/2, then we choose the root $p_{0}=\frac{3-\sqrt{3}}{6}$. Hence,*

$$F\left(p\right)⩽F\left(p_{0}\right)=\frac{\sqrt{3}}{18}\approx 0.0962.$$

*Thus the difference of the upper bound and the lower bound is about 0.1283 at most, and tends to 0 when n→+∞.*


**Example** **5.**
*Using the bounds in Theorem 7, the results in Figure 1 can be improved. See Figure 3.*

*When m=5, the bounds Equations *(Equation 15)* and *(Equation 14)* are also valid. See Figure 4.*

*Note that the difference of the bounds Equations *(Equation 15)* and *(Equation 14)* is about 0.05, which is much smaller than the given 0.1283.*


It is known that the Hamming codes satisfy the $2^{-m}$ bound when 0<p<1/2 i.e., $P_{ue}⩽2^{-m}$. See [5] for more details. In fact, the obtained new bound is better than the ordinary $2^{-m}$ bound, when *p* is not large.

**Theorem** **8.**
*Let $H_{m}$ be the binary $\left[n=2^{m}-1,k=n-m,3\right]$ Hamming code, then when 0<p<1/2 and m≥3, we have*

(16)
$$P_{ue}⩽\frac{p-p^{2}}{\left(n-1\right)p^{2}-5p+3}+p^{n}.$$

*Moreover, if $p<p_{0}$, this upper bound is better than the $2^{-m}$ bound, where $p_{0}$ is the smaller root of the equation $\left(2^{m+1}-2\right)x^{2}-\left(2^{m}+5\right)x+3=0$.*


**Proof.** Assume that
$$\frac{p-p^{2}}{\left(n-1\right)p^{2}-5p+3}<\frac{1}{2^{m}},$$
then it is sufficient to solve the inequality
$$\left(2^{m+1}-2\right)p^{2}-\left(2^{m}+5\right)p+3>0.$$Obviously, the inequality holds when $p<p_{0}$, where
$$p_{0}=\frac{\left(2^{m}+5\right)-\sqrt{{\left(2^{m}+5\right)}^{2}-12\left(2^{m+1}-2\right)}}{2(2^{m+1}-2)}$$
is the smaller root of the equation $\left(2^{m+1}-2\right)x^{2}-\left(2^{m}+5\right)x+3=0$. □

**Example** **6.**
*It is clear that when p is small enough, the new upper bound Equation *(Equation 14)* is smaller than the $2^{-m}$ bound in Figure 3 and Figure 4.*


**Remark** **9.**
*Of course, the weight distribution of the binary Hamming codes can be computed and expressed by the sum of combinatorial numbers, which are usually very large when m is large. So, the method in this section is to estimate $P_{ue}$ quickly. Compared with the $2^{-m}$ bound, our bounds are better when p is small enough.*


## 5. Conclusions

In this paper, we studied the probability of an undetected error $P_{ue}$ and gave many bounds for $P_{ue}$. The main contributions of this paper are the following:(1)The bounds obtained from the linear programming problem are given in Theorem 4. The bounds obtained from the Polynomial Method are given. According to the main Theorem 5, we get Theorem 6 (applied to the codes with even distances) and Proposition 3.(2)Combining the results of [17], we give the bounds in Propositions 1 and 2.(3)We find sharper bounds for binary Hamming codes (see Theorems 7 and 8).

To the best of our knowledge, that is the very first time that the LP method has been applied to bound $P_{ue}$. Even though computing $P_{ue}$ exactly requires knowledge of the code weight spectrum, our bounds depend solely on the three parameters [n,k,d], of the code. The weight frequencies are only used as variables in the LP program. Knowing the three parameters [n,k,d] is the minimal requirement to use a code in applications.

To sum up, our bounds are most useful when the exact weight distribution is too hard to compute. Our bounds perform well when *p* is small enough and the kissing number $A_{d}$ is known, and there are many such codes.

We mention the following open problems. The readers interested in Hamming codes are suggested to derive bounds for general *q*-ary Hamming codes with q>2. Moreover, it is worth mentioning that the linear programming problem works better numerically than the Polynomial Method. The interest of the latter lies in producing bounds with closed formulas. It is a challenging open problem to derive better bounds with polynomials of degree higher than 2. 

## Figures and Tables

**Figure 1 entropy-25-01349-f001:**
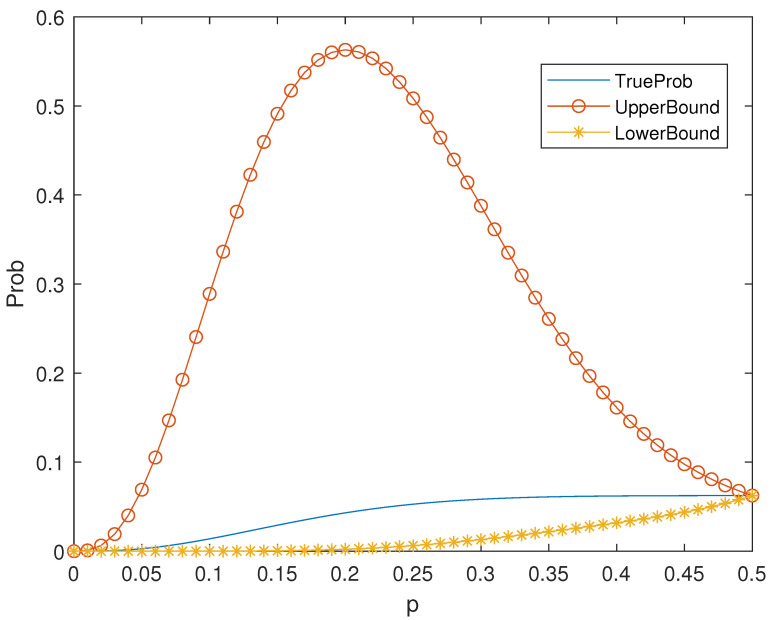
Bounds in Theorem 2 of $P_{ue}$ for the Hamming Code $H_{4}$.

**Figure 2 entropy-25-01349-f002:**
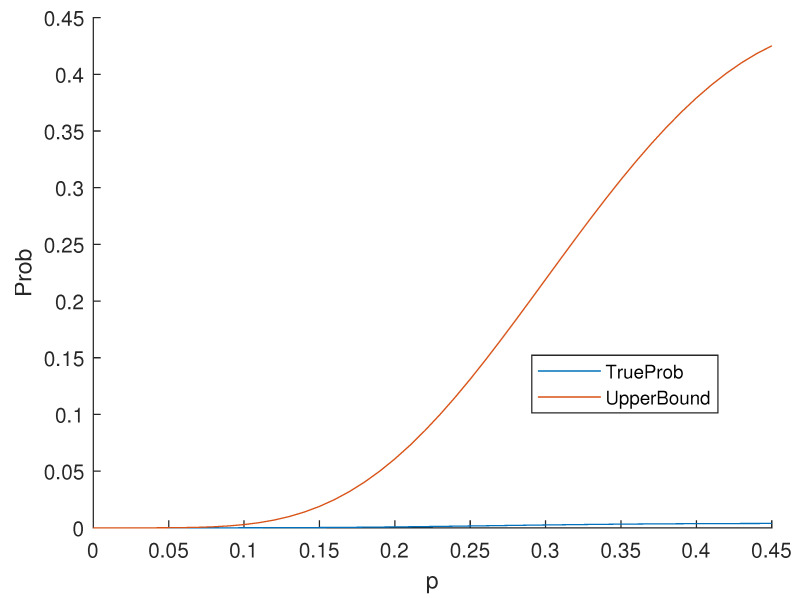
The Probability of Undetected Error of the Nordstrom–Robinson Code.

**Figure 3 entropy-25-01349-f003:**
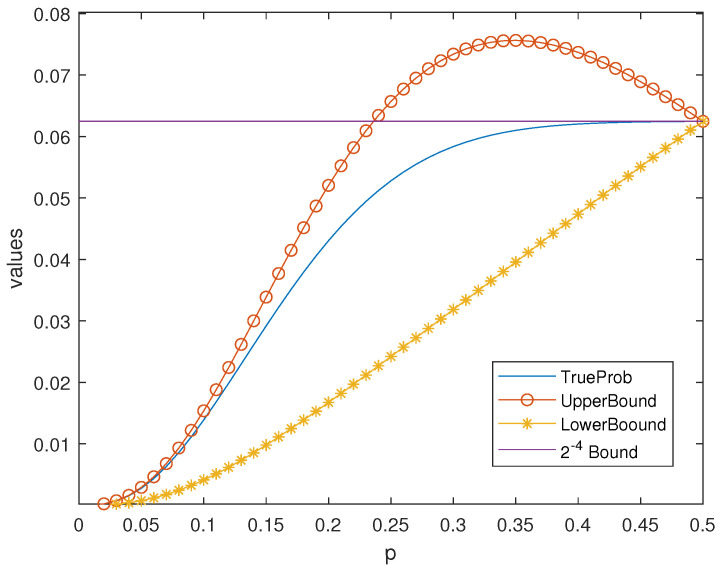
Bounds in Theorem 7 of $P_{ue}$ for the Hamming Code $H_{4}$.

**Figure 4 entropy-25-01349-f004:**
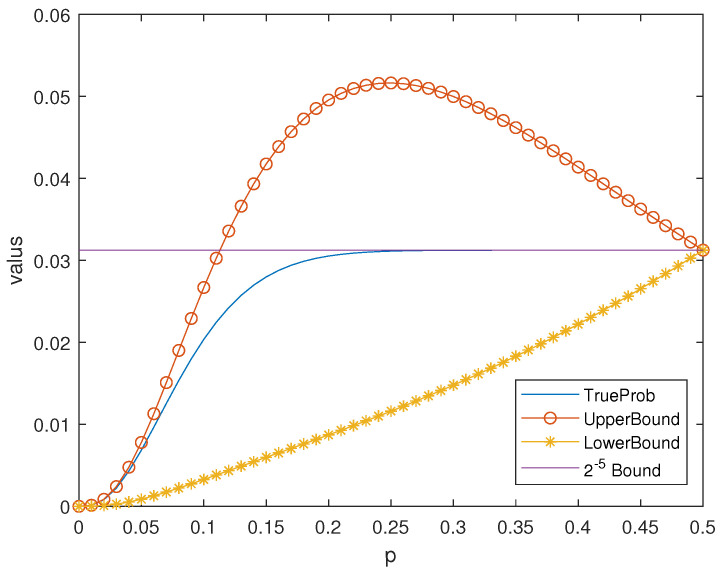
Bounds in Theorem 7 of $P_{ue}$ for the Hamming Code $H_{5}$.

**Table 1 entropy-25-01349-t001:** Distance Distribution of the Hamming Code $H_{4}$.

*i*	0	3	4	5	6	7	8	9	10	11	12	15
$A_{i}$	1	35	105	168	280	435	435	280	168	105	35	1

**Table 2 entropy-25-01349-t002:** Bounds of $A_{d}$ for Some Binary Codes.

Parameters	[9,4,4]	[10,4,4]	[11,4,5]	[12,4,6]	[13,4,6]	[14,4,7]	[15,4,8]
Upper Bound	14	15	7	14	14	8	15
Lower Bound	6	12	5	11	4	8	15

**Table 3 entropy-25-01349-t003:** Distance Distribution of the Nordstrom–Robinson Code.

*i*	0	6	8	10	16
$A_{i}$	1	112	30	112	1

## Data Availability

Data are available in a publicly accessible repository.

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
