# Peer review of "Bounds on the Probability of Undetected Error for q-Ary Codes"

_entropy, 2023, doi:10.3390/e25091349_

Round 1

Reviewer 1 Report

The authors present novel bounds for undetected error probability of linear codes.  The derivations appear to be valid, and the paper is well written.

A few minor remarks:

1. Please avoid using notation p', since it looks ugly and difficult to read when you write {p'}^j

2. "Linear programming program" -> "Linear program" or "Linear programming problem".

3. Please clarify the meaning of approximate inequalities which appear in Theorem 4 and Proposition 1

I congratulate the authors with nice results and suggest the paper to be accepted subject to correction of the above issues.

Author Response

3. Point-by-point response to Comments and Suggestions for Authors

Comments 1: Please avoid using notation p', since it looks ugly and difficult to read when you write {p'}^j

Response 1: Thank you for your suggestion. We will avoid the notation p' and use $p_q$.

Comments 2: "Linear programming program" -> "Linear program" or "Linear programming problem".

Response 2: Agree. We have corrected this expression.

Comments 3: Please clarify the meaning of approximate inequalities which appear in Theorem 4 and Proposition 1

Response 3: We add Remark 2 before Theorem 4.

Reviewer 2 Report

Overall the paper looks good. Two specific comments:

1. The background is not introduced thoroughly so that much extension reading is expected. For example, the two formulas below line 19 and 23 on page 1 are directly given (though references appear) and absolutely no explanation is available, making the work hard to read. Would be good to give some intuitive explanations/clues. 

2. On line 52 of page 2, the main result/approach of this work is described in a way that appears trivial - bounds on Ad have been studied and seems this work is using the same methods to study Pue; however, a direct relationship between Ad and Pd is known in (3), so are you applying the known bound Ad to (3) to obtain the desired bounds of Pue (this too naive and would indeed be the impression after reading to this part)?

N/A

Author Response

As for the "Can be improved" questions, please see the attachment. 

The following are just point-by-point responses.

Comments 1:  The background is not introduced thoroughly so that much extension reading is expected. For example, the two formulas below line 19 and 23 on page 1 are directly given (though references appear) and absolutely no explanation is available, making the work hard to read. Would be good to give some intuitive explanations/clues

Response 1: Thank you for your suggestion. We add more explanations from line 16 on page 1 to line 39 on page 2 in the revised version, which are too long to put them here.  

Comments 2: On line 52 of page 2, the main result/approach of this work is described in a way that appears trivial - bounds on Ad have been studied and seems this work is using the same methods to study Pue; however, a direct relationship between Ad and Pd is known in (3), so are you applying the known bound Ad to (3) to obtain the desired bounds of Pue (this too naive and would indeed be the impression after reading to this part)?

Response 2: Yes, we have applied the known bounds of Ad to (3), see Theorem 4, Proposition 3 and Proposition 2. Such results can be obtained directly, which is the motivation of this paper. Besides, we give some other novel bounds, and for these bounds, (3) is not used.

Reviewer 3 Report

In this paper, the authors studied bounds on the probability of undetected error for q-ary codes. I think the issue is very interesting. After reading the paper, I think that the paper is well written. However, I do not have time to  read the proofs carefully. It seems that all the results are correct. Thereby, I recommend its publication.

Author Response

Thank you for your recommendation! 
